# ACEK Biosensor for the Minute-Scale Quantification of Breast Cancer ctDNA

**DOI:** 10.3390/s24020547

**Published:** 2024-01-15

**Authors:** Ke Wang, Xiaogang Lin, Maoxiao Zhang, Mengjie Yang, Xiang Shi, Mingna Xie, Yang Luo

**Affiliations:** 1Key Laboratory of Optoelectronic Technology and Systems of Ministry of Education of China, Chongqing University, Chongqing 400044, China; kewang@stu.cqu.edu.cn (K.W.); 15223897701@163.com (M.Z.); 202208131095@stu.cqu.edu.cn (M.Y.); shixiangcqu98@163.com (X.S.); mingnaxie2017cqu@163.com (M.X.); 2Center of Smart Laboratory and Molecular Medicine, NHC Key Laboratory of Birth Defects and Reproductive Health, School of Medicine, Chongqing University, Chongqing 400044, China

**Keywords:** breast cancer, liquid biopsy, ctDNA, biosensor, PIK3CA E542K

## Abstract

Circulating tumor DNA (ctDNA) appears as a valuable liquid biopsy biomarker in the early diagnosis, treatment, and prognosis of cancer. Here, a biosensing method derived from the AC electrokinetics (ACEK) effect was constructed in this study for the simple, efficient, and rapid method of detection of ctDNA. In the proof-of-concept experiment, ctDNA from the PIK3CA E542K mutant in breast cancer was quantified by detecting a normalized capacitance change rate using a forked-finger gold electrode as the sensing electrode in combination with the ACEK effect. We compared two formats for the construction of the approach by employing varied immobilization strategies; one is to immobilize the DNA capture probe on the electrode surface by Au–S bonding, while the other immobilizes the probe on a self-assembled membrane on the electrode surface by amide bonding. Both formats demonstrated ultrafast detection speed by completing the ctDNA quantification within 1 min and a linear range of 10 fM–10 pM was observed. Meanwhile, the immobilization via the self-assembled membrane yielded improved stability, sensitivity, and specificity than its Au–S bonding counterpart. A detection limit of 1.94 fM was eventually achieved using the optimized approach. This research provides a label-free and minute-scale universal method for the detection of various malignant tumors. The ctDNA biosensors based on the ACEK effect improved according to the probe type or electrode structure and have potential applications in tumor drug efficacy prediction, drug resistance monitoring, screening of high-risk groups, differential diagnosis, monitoring of tiny residual lesions, and prognosis determination.

## 1. Introduction

Breast cancer is one of the most common cancers that threaten women’s lives [1]. In breast cancer patients, nearly 35% present with activating mutations in the PIK3CA gene, and 80–90% of these mutations are E542K, E545K, H1047L, and H1047R mutations [2]. Among them, PIK3CA E542K mutations are the most representative, and accurate detection of PIK3CA E542K mutations may significantly advance the accurate diagnosis and treatment of breast cancer development. A prerequisite for accurate diagnosis and treatment is accurate molecular typing of the breast cancer patient. Conventional molecular typing is based on the “gold standard” of tissue biopsy [3]. In clinical practice, this method has limitations such as the fact that it cannot be used for the early diagnosis of disease. It is not possible to repeat sampling at multiple time points due to the invasive nature of tissue acquisition, and the tumor heterogeneity makes the obtained tissue unrepresentative of the whole tumor [4]. Liquid biopsy is a non-invasive test based on samples of body fluids such as blood, urine, and saliva, which has the advantages of convenient sampling, repeatable sampling, and facilitates continuous monitoring [5]. The implementation of precision diagnosis and treatment of breast cancer relies on the effective application of highly sensitive and accurate biomarkers [6]. The main test markers of tumor liquid biopsy are circulating tumor cells (CTC), circulating tumor DNA (ctDNA), and exosomes [7,8,9]. Among these, ctDNA contains genetic material from all tumor clones and can reflect the full spectrum of tumor genetic information [10,11,12,13]. Figure 1a shows the process of generating and acquiring ctDNA for breast cancer. Dynamic monitoring of ctDNA allows accurate observation of the chronic response of tumors, which is of great value for both disease prognosis and targeted therapy [14].

However, ctDNA detection still faces many difficulties due to its short half-life and very small percentage of total cell-free DNA (cfDNA) [15,16]. Although technologies such as digital PCR (dPCR), BEAMing, tagged-amplicon deep sequencing (TAM-Seq), cancer personalized profiling by deep sequencing (CAPP-Seq), and targeted error correction sequencing (TEC-Seq) have made a significant contribution to the detection of ctDNA, further application of most of these methods is still hampered by cumbersome steps, expensive instruments, and long detection times [17,18,19,20,21]. Therefore, it is necessary to develop a simple, sensitive, specific, and low-cost method for the rapid detection of ctDNA.

In recent years, there has been a proliferation of electrochemical, optical, and piezoelectric biosensors for ctDNA detection [22,23,24,25,26]. Electrochemical biosensors have become a research hotspot for tumor ctDNA detection because of their good biocompatibility, specificity, high sensitivity, and applicability for the development of microdevices. Electrochemical biosensors use electrodes as conversion elements to convert the captured response signal into voltage, current, or resistance [27]. The main electrochemical methods commonly used to detect ctDNA are electrochemical impedance spectroscopy (EIS) and differential pulse voltammetry (DPV) [28]. The relative molecular mass of PIK3CA E542K mutant ctDNA is small, and its content in actual samples is low. In the detection process based on these methods, the binding of ctDNA to the probe relies on the free diffusion of the ctDNA molecule and the attraction of base complementary pairing, resulting in a long detection time. Therefore, it is difficult to ensure the activity of ctDNA and probe molecules, which is not conducive to real-time dynamic detection.

The AC electrokinetics (ACEK) effect is a group of microfluidic phenomena generated under the action of a non-uniform alternating electric field that can synchronize the signal reading of the biosensor and the enrichment of the particles, realizing the integration of enrichment and detection. The ACEK effect mainly includes the dielectrophorisis (DEP) effect, the AC electroosmosis (ACEO) effect, and the AC electrothermal (ACET) effect [29,30]. Depending on the difference between the particle and solution polarization rates, the DEP can move the particles to the high-field region (positive DEP) or the low-field region (negative DEP), resulting in particle enrichment [31,32]. In a non-uniform AC electric field, the electric field tangential to the electrode surface will drive the directional movement of charges in the electrical double layer (EDL), resulting in eddy electroosmosis and the formation of the ACEO effect [33]. Due to the continuity of the current and Gauss's law, the temperature gradient generated by the non-uniform electric field in the electrolyte solution will lead to changes in the dielectric constant and conductivity of the solution, along with the generation of mobile charges. The directional motion of the moving charge leads to a directional motion of the viscous fluid, resulting in the ACET effect [34]. The non-uniform AC electric field can generate micro-currents through the ACEO and ACET mechanisms or directly apply forces to the target material through the DEP mechanism, thereby transporting the target material to the electrode surface [35,36,37]. It has been shown that the ACEK forces of particles in solution are much larger than natural forces such as Brownian motion diffusion [38]. Therefore, the introduction of the ACEK technique in biosensors can accelerate the movement and enrichment of molecules in solution [39]. The ACEK sensor schematic is shown in Figure 1b. Since the ACET effect is not affected by the background solution conductivity or AC signal frequency, it dominates the testing process in the research of this paper.

Herein, a capacitive biosensor for ctDNA detection of the PIK3CA E542K mutant was constructed based on ACEK capacitive sensing technology, and the way of immobilizing the probe was optimized. During the detection process, the ACET effect is generated when an AC signal of fixed voltage and frequency is applied to the electrode, causing the solution carrying breast cancer ctDNA to move in a directional manner toward the surface of the electrode, where the capture probe is immobilized. The complementary pairing of ctDNA and probe bases will lead to an increase in the surface area of the interfacial capacitance between the electrode and the bilayer formed by the solution, which in turn increases the interfacial capacitance. Since the amount of bound ctDNA is positively correlated with the amount of change in interfacial capacitance, monitoring the change in interfacial capacitance during the detection process enables the quantitative detection of ctDNA in breast cancer. The proposed liquid biopsy method, based on ACEK capacitive sensing technology, accelerates the binding of capture probes to ctDNA and shows good sensitivity and specificity. It demonstrates great potential for accurate cancer diagnosis and treatment.

## 2. Experimental Section

### 2.1. Materials and Reagents

Phosphate buffer (1 × PBS, 137 mM NaCl, 2.7 mM KCl, and 10 mM phosphate, pH 7.2–7.4) purchased from MP Biomedicals (USA) was used as the background solution for ctDNA in this experiment. Isopropanol, anhydrous ethanol, 6-mercapto-1-hexanol (6-MCH), and 1-ethyl-(3-dimethylaminopropyl) carbodiimide hydrochloride (EDC) were purchased from Shanghai Maclean (China). Acetone was purchased from Chengdu Kolon (China). ctDNA oligonucleotides were synthesized by Shanghai Sangon (China). The sequences of the capture probe DNA and breast cancer ctDNA are listed in Table 1. In addition, a wild-type DNA strand and a fully non-complementary DNA strand were set up, where the wild-type DNA strand differs from the ctDNA strand by only one base and the fully non-complementary DNA strand differs from the ctDNA strand by all bases. Other experimental materials and chemical reagents used in this study were obtained from standard reagent suppliers.

### 2.2. Chips Pre-Processing

As the aspect ratio of the fork-finger electrodes is larger and the gap is smaller, the sensitivity and response speed of the sensor will be better. Therefore, 5 µm × 5 µm chips were selected for subsequent experiments. The chip needs to be cleaned before modifying the electrode. Firstly, the forked-finger gold electrode chip was soaked in acetone solution for 24 h to remove the photoresist left on the surface during the chip processing. Then the chip was removed, rinsed with ultrapure water for 30 s, and blown dry with nitrogen gas. The blow-dried chip was soaked again in isopropyl alcohol for 5 min. After removing the chip, it was rinsed with anhydrous ethanol for 30 s, rinsed with ultrapure water for 30 s, and then blown dry with nitrogen gas. Then, a multimeter was used to measure the chip resistance value. If the resistance value exceeded 200 MΩ, the chip surface was cleaned; otherwise, the above steps were repeated.

### 2.3. Surface Modification and Hybridization

The chip was placed in the UV ozone cleaner for 20 min before the modification of the probe to increase the oxygen negative ion content on the electrode surface, thus increasing the hydrophilicity and enabling the probe to be better adsorbed on the gold electrode surface. In this paper, two methods were used to immobilize the probe, as shown in Figure 1c. One is to immobilize the probe by Au–S bonding, which is incubated by adding 20 μL of sulfhydryl-modified probe solution dropwise directly on the electrode surface. After the incubation of the probe was completed, 10 µL of 6-MCH solution was added dropwise to close the sites not bound to the probe and prevent impurities from binding non-specifically to the electrode during ctDNA detection. The other is to immobilize the probe by amide bonding. Before immobilizing the probe, the chip needs to be immersed in MUA solution to form a dense MUA self-assembly film on the electrode surface. Then EDC and NHS solution were mixed in a ratio of 1:4 by volume to activate the carboxyl group at one end of the self-assembled membrane. The activated self-assembled membrane can be bound to the amino-modified probe via amide bonding. After the probe was fixed, 10 µL of ethanolamine solution was added dropwise to the lumen as a closure solution to prevent non-specific binding during the assay. After these steps, two specifically functionalized ctDNA biosensors were prepared. Finally, a specific AC electrical signal was applied at both ends of the electrodes, and then ctDNA solution was added dropwise to the electrode surface with the probe fixed on the electrode surface for quantitative detection.

### 2.4. Characterization of ACEK Biosensors

The sensitivity and specificity of the ACEK biosensor were characterized using an IM3536 LCR tester (Hioki, Shanghai). The AC signal was applied to the electrode by the IM3536 LCR tester, and the interfacial capacitance on the electrode surface was monitored at the same time. The process of modifying the electrode was characterized using a Thermo Fisher K-Alpha+ X-ray photoelectron spectrometer (XPS) and scanning electron microscopy (SEM).

### 2.5. Electrochemical Measurements

The system for electrochemical measurements on ctDNA samples is shown in Figure 2. The measurement frequency of the IM3536 LCR tester is 4 Hz–8 MHz. Through experimental testing, the excitation voltage was set to 300 mV at 20 kHz, and the best ACEK results were produced at this excitation. The measurement method was set to time scan, with a continuous measurement time of 1 min and a scan interval of 1 s. To reduce the effect of experimental chance errors, each measurement was repeated three times.

## 3. Results and Discussion

### 3.1. Surface Properties of the Functionalized Electrodes

In this paper, XPS was used to analyze the changes in the chip surface before and after the Au–S bonding fixed capture probes; XPS and SEM were also used to analyze the changes in the chip surface before and after the formation of the MUA self-assembly film. The analysis results are shown in Figure 3. Figure 3a,b shows the XPS full spectra of the chip surface before and after the Au–S bonding fixed capture probes. Compared to before fixation of the probe, four elements of Na (from PBS), N (from aptamer), Cl (from PBS), and P (from probe) appeared after fixation of the probe. This provides evidence for the successful immobilization of the probe on the electrode surface. Figure 3c,d shows the complete XPS spectra of the chip surface before and after the formation of the self-assembled film, with the S element appearing only after the formation of the self-assembled film compared to before the formation of the self-assembled film. Figure 3e,f shows the SEM images of the electrode surface before and after the self-assembled film formation. The electrode surface is smooth before the self-assembled film formation, and the roughness of the electrode surface increases significantly after the formation of the self-assembled film.

### 3.2. Working Principle of the ACEK Biosensor

In this study, a silicon-substrate fork-finger gold electrode was used as a biosensing chip because gold has better conductivity and stability, and the fork-finger electrode can better produce the ACEK effect, which facilitates the immobilization of more probes and enhances the strength of the electrochemical signal. A chamber with a diameter of 2 mm and a depth of 0.9 mm was affixed to the chip, which not only allows precise incubation of the probes in the forked finger section but also reduces the sample volume and lowers the cost. Upon application of an AC electrical signal, the ACEK effect induces rapid enrichment of PIK3CA E542K ctDNA molecules to the electrode surface and specific binding to the designed DNA probe. The biosensor converts the biological information generated by the specific binding into a measurable electrical signal through the sensing electrode, and information on the concentration of ctDNA can be analyzed based on the measured capacitive signal change rate.

### 3.3. Characterization of the Dominant Effect in ACEK

To evaluate the effectiveness of DEP, ACEO, and ACET effects during ctDNA detection, the conductivity of the background solution 0.05 × PBS was measured using a conductivity meter, and the results are shown in Table 2. Since the conductivity of 0.05 × PBS is higher than 0.085 S/m, ACEO will fail during the assay, and only ACET or DEP may occur. By applying AC signals of 1 kHz and 20 kHz to this biosensor, the detection results are less affected by the frequency change (Appendix A). Therefore, the ACET effect dominates in this study.

### 3.4. Detection and Sensitivity of the ACEK Biosensor

To obtain the best detection performance of the sensor, a chip with a size of 5 µm × 5 µm was chosen, and the frequency and voltage of the applied AC signal were set at 20 kHz and 300 mV, respectively. Briefly, 10 fM, 100 fM, 1 pM, and 10 pM ctDNA solution and 0.05 × PBS were detected, and 0.05 × PBS was used as a blank control. The detection results of the method of Au–S bonding immobilized probes and the method of amide bonding immobilized probes are shown in Figure 4. Figure 4a,c shows the curves of normalized capacitance with time for the two methods detecting four concentrations of ctDNA solution. The normalized capacitance increases with time over 1 min. In addition, the normalized capacitance increases with increasing concentrations of ctDNA solution. It means that the amount of ctDNA bound to the probe increases. Figure 4b,d shows the linear fit curves of the normalized capacitance rate of change versus concentration obtained from the two methods detecting four concentrations of ctDNA solution. In the concentration range of 10 fM–10 pM, the normalized capacitance rate of change y exhibited a logarithmic dependence on the solution concentration x.

The trend of the normalized interfacial capacitance obtained using the method of Au–S bonding immobilized probes over time is shown in Figure 4a, where it can be seen that the normalized interfacial capacitance increases with time and ctDNA concentration. Figure 4b shows the fitted relationship between the normalized interfacial capacitance rate of change and ${log}_{10}x$. The relationship between the normalized interfacial capacitance rate of change and the ctDNA concentration obtained by the method of Au–S bonding immobilized probes is shown in Equation (1):(1)$$y_{1}=8.303\log_{10}(x)-6.319(R^{2}=0.984)$$
where $y_{1}$ is the normalized interfacial capacitance rate of change, with a Pearson correlation coefficient of $R^{2}=0.984$. The limit of detection (LOD) of the sensor was determined by subtracting 3 standard deviations (3σ) from the response of the blank control sample [40]. The response of the background solution 0.05 × PBS was 0.02 ± 0.02%/min, from which the cut-off normalized interfacial capacitance change rate was calculated to be 0.08%/min. Substituting this into Equation (1) yields a detection limit of 5.90 fM for the biosensor using the method of Au–S bonding immobilized probes.

The capacitance response obtained by the method of amide bonding immobilized probes is shown in Figure 4c, where the normalized interfacial capacitance increases with time and ctDNA concentration. Figure 4d shows the fitting curve of the normalized interfacial capacitance rate of change versus ctDNA concentration. The fitting relationship is shown in Equation (2):(2)$$y_{2}=9.009\log_{10}(x)-2.327(R^{2}=0.992)$$
where $y_{2}$ is the normalized interfacial capacitance rate of change obtained by this method, with a Pearson correlation coefficient $R^{2}=0.992$. The detection limit of the biosensor was also calculated based on the response of the blank control sample minus 3σ. The response of the background solution 0.05 × PBS solution was −0.04 ± 0.10%/min, and the cut-off normalized interfacial capacitance rate of change was calculated to be 0.26%/min. Substituting this value into Equation (2) yields a limit of detection of 1.94 fM for the detection of ctDNA by the method of amide bonding immobilized probes.

Comparing the results of the two methods, the slope of the normalized capacitance change rate versus concentration straight line obtained by the method of amide bonding immobilized probes was higher. Due to the higher stability of the amide bond than the Au–S bond, this means that more of the probe was immobilized on the self-assembled membrane on the electrode surface through the amide bond, and therefore was able to bind more ctDNA. In addition, the Pearson correlation coefficient obtained by this method is also greater, indicating that the method is closer to the true value and more accurate for the quantitative detection of ctDNA. Compared with other electrochemical methods for the detection of PIK3CA ctDNA (Appendix A), the proposed biosensor exhibited excellent specificity and sensitivity with a relatively short detection time. The biosensor is expected to enable point-of-care testing (POCT) in the early stages of breast cancer, further highlighting the analytical potential of the biosensor in clinical applications.

### 3.5. Specificity of the ACEK Biosensor

The specificity of two biodetectors for breast cancer ctDNA was evaluated by comparing the rate of capacitance change of ctDNA samples using two biosensing strategies. Wild-type DNA solution and completely non-complementary DNA solution at concentrations of 100 fM, 1 pM, and 10 pM were detected by two methods, with 0.05 × PBS as a blank control. The test results are shown in Figure 5. The normalized capacitance change rates of even wild-type DNA solution and fully non-complementary DNA solution with concentrations up to 10 pM are much smaller than those of ctDNA solution with 100 fM. The rate of normalized capacitance change was much smaller than that of the ctDNA solution at 100 fM. Therefore, the biosensor is highly specific for ctDNA, and its non-specific adsorption is negligible.

### 3.6. Repeatability

The reproducibility of the biosensor was verified by three different biosensors using the method of amide bonding immobilized probes, and three detection experiments were performed on 10 pM ctDNA samples with 0.05 × PBS as background solution. The normalized capacitance values obtained from the experiments are shown in Figure 6a, and the normalized capacitance change rates for the three cycles were 33.73%, 34.20%, and 32.47%, respectively (Figure 6b). The biosensor has excellent reproducibility, and the difference in response between the three biosensors was within 1.73%.

## 4. Conclusions

The ACEK capacitive sensing technology-based biosensor proposed in this paper achieves rapid, low-cost, highly sensitive, and highly specific liquid biopsy of PIK3CA E542K ctDNA. Thiol-modified and amino-modified specific probes were immobilized on a forked-finger gold electrode, and an AC signal was applied to the electrode to induce the ACEK effect, which accelerated the binding of ctDNA to the probe and measured the change in interfacial capacitance on the electrode surface. The experimental results showed that the prepared biosensor could be detected in only 1 min with good linearity in the ctDNA concentration range of 10 fM–10 pM. The detection limit obtained by the method of Au–S bonding immobilized probes was 5.90 fM, and that by the method of amide bonding immobilized probes was only 1.94 fM with higher detection sensitivity. It was also shown that the slope of the straight line about normalized capacitance change rate versus concentration change obtained by the method of amide bonding immobilized probes is higher. This means that more ctDNA binds to the probe per unit time, indicating that amide bonding is more conducive to probe immobilization. Thus, this liquid biopsy strategy provides a feasible solution for ctDNA detection and can further promote the development of precision medicine. In future studies, we will explore the response recovery characteristics of this biosensor and use this ctDNA biosensor for clinical sample detection to provide a reliable clinical basis for the application of this biosensor. In addition, by improving the electrode into an array electrode, the simultaneous detection of multiple ctDNA mutations can be realized.

## Figures and Tables

**Figure 1 sensors-24-00547-f001:**
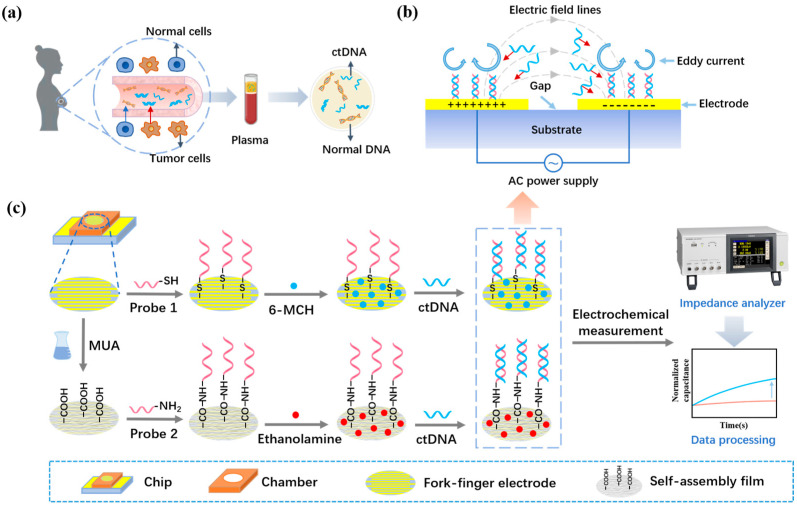
(**a**) The process of ctDNA production and acquisition in breast cancer patients. (**b**) The AC signal excites the ACEK effect, accelerating the movement of biomolecules between electrodes. (**c**) Sensing strategies for a rapid liquid biopsy biosensor of breast cancer ctDNA.

**Figure 2 sensors-24-00547-f002:**
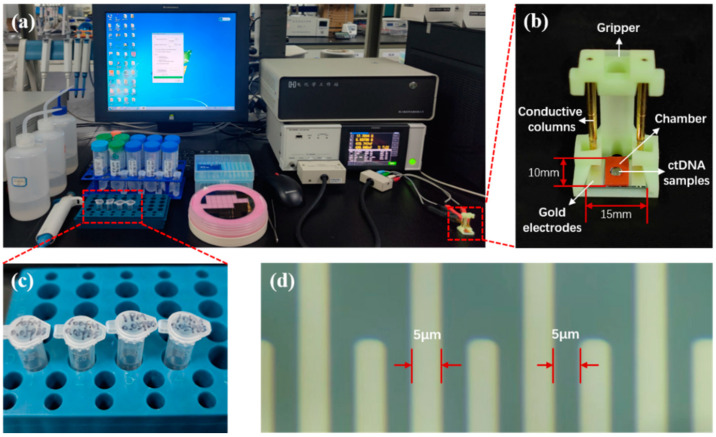
(**a**) Equipment and reagents for ctDNA electrochemical measurements. (**b**) ACEK biosensor and its fixation device. (**c**) 10 fM, 100 fM, 1 pM, and 10 pM ctDNA samples. (**d**) Forked-finger gold electrodes under the microscope.

**Figure 3 sensors-24-00547-f003:**
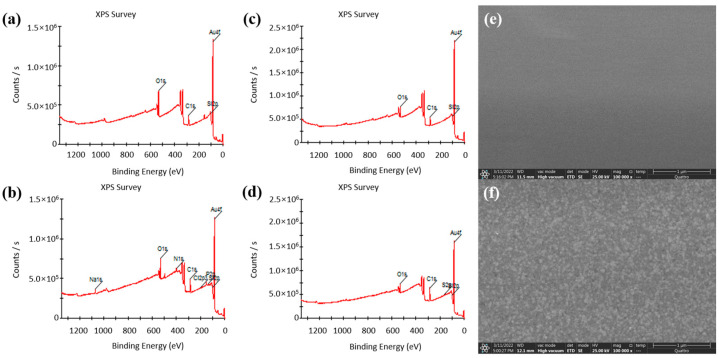
(**a**) XPS full spectrum of the chip surface before the probe was fixed by Au–S bonding. (**b**) XPS full spectrum of the chip surface after the probe was fixed by Au–S bonding. (**c**) XPS full spectrum of the chip surface before the formation of self-assembled film. (**d**) XPS full spectrum of the chip surface after the formation of the self-assembled film. (**e**) SEM image of the electrode surface before the formation of the self-assembled film. (**f**) SEM image of the electrode surface after the formation of the self-assembled film.

**Figure 4 sensors-24-00547-f004:**
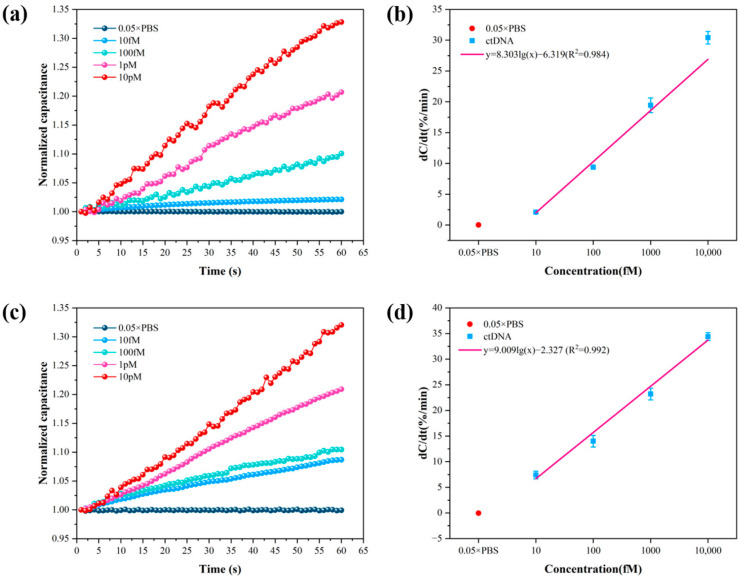
(**a**) Normalized capacitance change rate versus time for 4 concentrations of ctDNA solutions detected using the method of Au–S bonding immobilized probes. (**b**) Normalized capacitance change rate versus concentration for 4 concentrations of ctDNA solutions using the method of Au–S bonding immobilized probes. (**c**) Normalized capacitance change rate versus time for 4 concentrations of ctDNA solutions using the method of amide bonding immobilized probes. (**d**) Detection of normalized capacitance change rate versus concentration for 4 concentrations of ctDNA solutions using the method of amide bonding immobilized probes.

**Figure 5 sensors-24-00547-f005:**
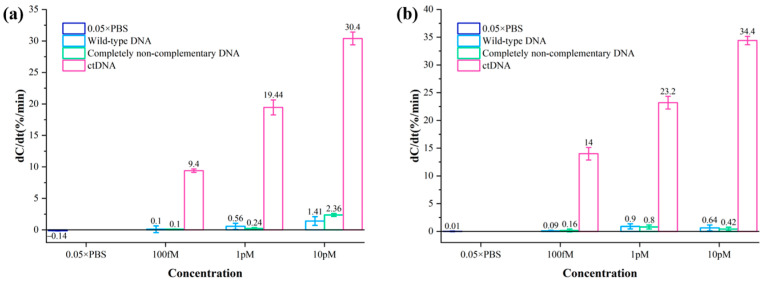
(**a**) Normalized capacitance change rates of different concentrations of wild-type DNA solutions and completely non-complementary DNA solutions using the method of Au–S bonding immobilized probes. (**b**) Normalized capacitance change rates of different concentrations of wild-type DNA solutions and completely non-complementary DNA solutions using the method of amide bonding immobilized probes.

**Figure 6 sensors-24-00547-f006:**
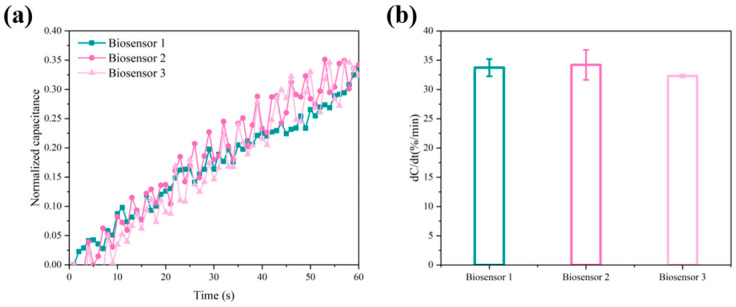
(**a**) The normalized capacitance versus time curve for the biosensor over three cycles. (**b**) The normalized capacitance change rates of the biosensor over three cycles.

**Table 1 sensors-24-00547-t001:** Nucleotide sequences used in the experiments.

Nucleic Acid	Sequences (5′-3′)
Thiol-modified probe	HS-C6-AGTGATTTCAGAGAG
Amino-modified probe	NH2-C6-AGTGATTTCAGAGAG
ctDNA	AACAGCTCAAAGCAATTTCTACACGAGATCCTCTCTCTGAAATCACTGAGCAGGAGAAAGATTTTCTATGGAGTC
Wild-type DNA	AACAGCTCAAAGCAATTTCTACACGAGATCCTCTCTCTAAAATCACTGAGCAGGAGAAAGATTTTCTATGGAGTC
Completely non-complementary DNA	AGATCCAATCCATTTTTGTTGTCCAGCCACCATGATGCGCATCATTCATTTGTTTCATGAAATACTCC

**Table 2 sensors-24-00547-t002:** Test results of conductivity of 0.05 × PBS solution.

Sample	Temperature/°C	Test Value 1	Test Value 2	Test Value 3	Average Value
0.05 × PBS	18.9 °C	0.0988 S/m	0.0988 S/m	0.0989 S/m	0.0988 S/m

## Data Availability

Data are contained within the article and Appendix A.

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
