# Peer review of "ACEK Biosensor for the Minute-Scale Quantification of Breast Cancer ctDNA"

_sensors, 2024, doi:10.3390/s24020547_

Round 1

Reviewer 1 Report

Comments and Suggestions for Authors

The paper titled "ACEK biosensor for the minute-scale quantification of breast cancer ctDNA" outlines a promising approach for rapid and sensitive detection of breast cancer circulating tumor DNA (ctDNA) using AC electrokinetics (ACEK) technology. Please accept my congratulations on the beautiful study. I have a few suggestions

Suggestions regarding the Abstract:

Strengths: Clear and concise writing: The abstract effectively summarizes the key aspects of the research, including the motivation, method, results, and potential applications. The novel approach and the use of the ACEK effect for rapid ctDNA detection appears to be a new and promising with potential advantages over existing methods. The reported detection limit of 1.94 fM is promising and suggests high sensitivity for ctDNA detection.

Areas for improvement: Specificity: The abstract mentions improved specificity with the self-assembled membrane approach, but it would be helpful to provide more details about how this was achieved and whether any cross-reactivity with non-target DNA was observed.

Clinical relevance: The abstract could benefit from a more specific discussion of how this technology could be translated into a practical diagnostic tool.

Conclusion: The abstract mentions the potential for wider applications, but it would be good to provide a more concise and specific concluding statement that summarizes the key findings and their significance.

For the entire paper:

Areas for further discussion: Comparison with existing methods: It would be helpful to compare the performance of the ACEK biosensor with established ctDNA detection methods in terms of sensitivity, specificity, and cost.

Clinical validation: While the results using cell lines are promising, further validation using clinical samples is crucial for assessing the real-world efficacy of the biosensor.

Multiplexing capabilities: Exploring the possibility of adapting the biosensor for simultaneous detection of multiple ctDNA mutations could enhance its clinical utility.

Long-term stability: Investigating the long-term stability of the biosensor and its performance over time is essential for practical applications.

Comments on the Quality of English Language

Seems okay

Reviewer 2 Report

Comments and Suggestions for Authors

This manuscript proposed a biosensing method based on the ACEK effect for the detection of ctDNA from breast cancer. Two different immobilization strategies of the forked-finger sensing electrodes, Au–S bonding and amide bonding, were performed and compared. Ultrafast detection of ctDNA was achieved with an LOD of fM level. Overall, this work is interesting and showed potential practical significance, especially in clinical applications, and may provide a broad reading interest to the journal readers. However, some issues including the primary innovation of the method and showing of the Figures should be concerned. Therefore, I think this manuscript can be accepted for publication after several revisions.

Detailed comments listed below:

1.      The authors have not sufficiently explained the electrical working principle of the ACEK in this biosensor.

2.      The resolution of Figure 3 should be improved to clearly show the details of the XPS spectrum.

3.      Figure 4 is missed and cannot be find in the manuscript.

4.      What is the time response and response-recovery properties of the biosensor?

Reviewer 3 Report

Comments and Suggestions for Authors

The article is well-written, and the research presents a significant contribution to the field of cancer diagnostics. The inclusion of experimental details, surface analyses, and specificity assessments adds depth to the study.

Some suggestions and comments about this review paper are: while the article mentions the ACEK effect, it would be beneficial to provide a brief explanation or schematic diagram illustrating the three components: dielectrophoresis (DEP), AC electroosmosis (ACEO), and AC electrothermal (ACET) effects.

Discussing potential clinical applications and the translational potential of the biosensor in real-world scenarios would further emphasize its significance.

Providing a brief comparison with existing electrochemical methods for the detection of PIK3CA ctDNA (Table S1) could enhance the article's context and highlight the biosensor's advantages.

Reviewer 4 Report

Comments and Suggestions for Authors

This study introduces a simple, efficient, and fast biosensing technique based on ACEK) to detect ctDNA. The method was tested by quantifying ctDNA in breast cancer using a specialized gold electrode and comparing two immobilization strategies: Au–S bonding and amide bonding on a self-assembled membrane. The current manuscript is unsuitable for publication in terms of both information and experimental results. It is recommended that the authors undertake a comprehensive revision before resubmitting. Below are the main comments

1) In Figure 1, the image labeled C needs to be corrected. The author should define and refine this diagram to ensure that readers can clearly obtain information when reading it. The current schematic is missing many details, including the distinction between two types of fixtures and the working principles, among others.

2) The authors have employed XPS to characterize the differences in surfaces before and after treatment. However, the results and subsequent discussion are perplexing and lack persuasiveness. Firstly, for a and b, the elements the authors identified may simply be contaminants from the buffer solution. As for c and d, there are noticeable sulfur peaks in the spectra that the authors claim is absent, yet the peak intensities are not convincing enough. The authors should consider employing alternative methods like NMR to characterize the Au-S bonding.

3)  There is no Figure 4 in the manuscript.

4) The experiment lacks a reference sample (negative) for comparison

Comments on the Quality of English Language

overall writing proficiency is satisfactory, but there is room for improvement

Round 2

Reviewer 4 Report

Comments and Suggestions for Authors

The authors have done a comprehensive revision, so I think this work is

ready for publication